# Nanocrystalline Skinnerite (Cu_3_SbS_3_) Prepared by High-Energy Milling in a Laboratory and an Industrial Mill and Its Optical and Optoelectrical Properties

**DOI:** 10.3390/molecules28010326

**Published:** 2022-12-31

**Authors:** Erika Dutková, María Jesús Sayagués, Martin Fabián, Matej Baláž, Jaroslav Kováč, Jaroslav Kováč Junior, Martin Stahorský, Marcela Achimovičová, Zdenka Lukáčová Bujňáková

**Affiliations:** 1Institute of Geotechnics, Slovak Academy of Sciences, 04001 Košice, Slovakia; 2Institute of Material Science of Seville (CSIC-US), Avenida Américo Vespucio 49, 41092 Seville, Spain; 3Institute of Electronics and Photonics, Slovak University of Technology, 81219 Bratislava, Slovakia

**Keywords:** mechanochemistry, milling, ternary chalcogenide Cu_3_SbS_3_, nanocrystals, optical properties, optoelectrical properties

## Abstract

Copper, antimony and sulfur in elemental form were applied for one-pot solid-state mechanochemical synthesis of skinnerite (Cu_3_SbS_3_) in a laboratory mill and an industrial mill. This synthesis was completed after 30 min of milling in the laboratory mill and 120 min in the industrial mill, as corroborated by X-ray diffraction. XRD analysis confirmed the presence of pure monoclinic skinnerite prepared in the laboratory mill and around 76% monoclinic skinnerite, with the secondary phases famatinite (Cu_3_SbS_4_; 15%), and tetrahedrite (Cu_11.4_Sb_4_S_13_; 8%), synthesized in the industrial mill. The nanocrystals were agglomerated into micrometer-sized grains in both cases. Both samples were nanocrystalline, as was confirmed with HRTEM. The optical band gap of the Cu_3_SbS_3_ prepared in the laboratory mill was determined to be 1.7 eV with UV–Vis spectroscopy. Photocurrent responses verified with I–V measurements under dark and light illumination and Cu_3_SbS_3_ nanocrystals showed ~45% enhancement of the photoresponsive current at a forward voltage of 0.6 V. The optical and optoelectrical properties of the skinnerite (Cu_3_SbS_3_) prepared via laboratory milling are interesting for photovoltaic applications.

## 1. Introduction

A wide variety of copper-based semiconducting chalcogenides have been investigated in recent years to pursue the need for solar-cell materials. Materials based on the Cu-Sb-S ternary system are regarded to be potential low-cost, sustainable absorbers capable of use in thin-film solar cells due to the availability and the low cost of the constituent nontoxic and earth-abundant elements. Their applications, such as in near-infrared, telecommunication and solar photovoltaic devices as well as thermoelectrics are also interesting [1,2].

Skinnerite (Cu_3_SbS_3_) belongs among the semiconductors that are currently studied very intensively. Cu_3_SbS_3_ is a ternary semiconductor with a direct bandgap value that ranges from 1.46 to 1.84 eV and has high absorption coefficients with values over 10^5^ cm^−1^ [3]. Cu_3_SbS_3_ is also a promising material for thermoelectrics [4], and confirmation of its potential for solar-energy conversion applications has also been approved [5]. Furthermore, it can be used as a photocatalyst material in treatment of wastewater that contains huge quantities of dyes [6]. Cu_3_SbS_3_ nanocrystals are also used in photothermal, photovoltaic and sensing applications [7].

Skinnerite (Cu_3_SbS_3_) nanocrystals have been synthesized via various techniques, such as, sputtering [5], chemical-bath deposition [6], the hot-injection method [7,8], thermal evaporation [9,10,11,12], the solvothermal method [13,14,15] and wet-chemical synthesis [16].

Cu_3_SbS_3_ thin films have been prepared via the chemical-bath-deposition method in an aqueous solution that contained (CuCl_2_, 2H_2_O), SbCl_3_ and Na_2_S_2_O_3_ as precursors; these exhibit good photocatalytic activity to degrade methylene blue under solar and xenon-lamp irradiation [6]. Cu_3_SbS_3_ NCs with high phase purity have been synthesized through a hot-injection method [7]. Ikeda et al. [8] also synthesized homogeneous Cu-Sb-S nanocrystals with p-type semiconductive properties in a solution through the hot-injection method. Fabrication of Cu_3_SbS_3_ thin films using successive thermal evaporation of Cu_2_S and Sb_2_S_3_ layers was reported in paper [9]. Cu_3_SbS_3_ thin films prepared with the conventional thermal-evaporation technique used for fabrication of solar cells based on *n*-Si substrates were studied in [10]. Cu_3_SbS_3_ thin films with p-conductivity were synthesized via the single-source vacuum-thermal-evaporation method [11]. The annealing-temperature effect of Cu_3_SbS_3_ thin films prepared with the vacuum-evaporation technique was investigated in paper [12]. Cu_3_SbS_3_ nanowires were obtained through the mild solvothermal route based on reactions between CuCl_2_, SbCl_3_ and elemental sulfur in ethylenediamine [13]. Zhong et al. synthesized Cu_3_SbS_3_ nanorods via a simple and convenient biomolecule-assisted solvothermal route [14]. Cu_3_SbS_3_ nanocrystallites were also successfully acquired through a facile solvothermal route based on the reactions between CuCl_2_, SbCl_3_ and thioglycolic acid in glycol [15]. Atri et al. [16] investigated co-thermal decompositions of equimolar concentrations of thiourea complexes ([Cu(tu)_3_]Cl and [Sb(tu)_2_]Cl_3_) in ethylene glycol and ethanolamine.

The ball-milling method affords all of the advantages of the enlarging field of mechanochemistry [17]. These benefits include green, solvent-free synthesis of both organic [18] and inorganic materials [19] and top-down synthesis of nanoparticles [20]. This method represents an environmentally friendly alternative to the classical methods of nanoparticle synthesis. However, sometimes, a very long milling time or high milling speed is necessary. In most cases, a subsequent annealing step is utilized in order to obtain desired structures. Mechanochemistry can be upscaled from a laboratory to a hundred-gram or even pilot-plant scale. An eccentric vibratory mill can be used for larger scale production [21,22,23,24,25]. Synthesis of skinnerite (Cu_3_SbS_3_) using mechanical alloying (MA) as a solid-state route from elemental powders at 450 rpm for 20 h and using the hot-pressing and spark-plasma-sintering methods has already been performed [26].

Herein, a mechanochemical synthesis of skinnerite (Cu_3_SbS_3_) nanocrystals from elemental precursors, for a shorter milling time, without the following postheating via high-energy milling on a laboratory scale and with utilization of an argon atmosphere, is investigated. Moreover, the optical and optoelectrical properties of pure skinnerite that was mechanochemically synthesized in a planetary laboratory mill were also studied. The novelty thereof is mainly synthesis of skinnerite on the industrial scale.

## 2. Results

### 2.1. Reaction Kinetics and Structural Characterization of Skinnerite (Cu_3_SbS_3_) Mechanochemically Synthesized in a Laboratory and an Industrial Mill

A series of experiments were performed in a laboratory mill as well as in an industrial mill. In order to investigate the reaction kinetics in the laboratory-scale planetary ball mill, XRD patterns were recorded at different milling times (5, 15 and 30 min). They are presented in Figure 1, below.

The skinnerite (Cu_3_SbS_3_) (JCPDS 01-082-0851), was already identified at the shortest milling time (5 min), and an intermediate phase of covellite (CuS) (JCPDS 00-024-0060), was observed. Unreacted antimony (Sb) (JCPDS 01-076-8600), was also present. After 15 min of milling, only skinnerite (Cu_3_SbS_3_) (JCPDS 01-082-0851), was present as the final product, and the intermediate phases disappeared. The mechanochemical reaction was almost completed after 15 min of milling. No other phases were detected, indicating the high purity of the skinnerite product. A further 15 min of treatment brought about no significant changes in the XRD pattern. The crystal structure of skinnerite was monoclinic, with a space group of P121/c1. These results are consistent with previous reports in the literature [9].

The XRD patterns of the synthesized skinnerite (Cu_3_SbS_3_) for 60 and 120 min of milling in the industrial mill are shown in Figure 2.

In the case of the sample prepared with 60 min of milling, the peaks could be assigned to the monoclinic skinnerite (Cu_3_SbS_3_) phase (according to Rietveld refinement, its content was 38%) (JCPDS 01-082-0851). However, a few lower-intensity peaks corresponded to the secondary tetragonal famatinite (Cu_3_SbS_4_; 28%) (JCPDS 01-071-3305) and cubic tetrahedrite (Cu_11.4_Sb_4_S_13_; 31%) phases (JCPDS 01-075-2211) that overlapped with the skinnerite phase. Unreacted antimony (Sb; below 3%) (JCPDS 01-076-8600) was also present. Formation of famatinite as a secondary phase has also been reported in previous works [3,9]. The progress of the mechanochemical synthesis in the industrial mill was significantly improved after 120 min of milling with monoclinic skinnerite (Cu_3_SbS_3_) in an amount of approximately 76%. Through a milling-time increase to 120 min, a smaller amount of the famatinite phase (Cu_3_SbS_4_; 15%), as well as of the tetrahedrite phase (Cu_11.4_Sb_4_S_13_; 8%), were evidenced. However, the unreacted antimony (Sb; below 1%) was also present. All of these XRD patterns were strongly influenced by amorphization and the generation of small crystallites.

Monoclinic skinnerite (Cu_3_SbS_3_) was formed as a final product in both cases, following Equation (1). From the kinetics investigations presented above, it was concluded that the best sample was the sample milled in the laboratory mill for 30 min. While the mechanochemical synthesis in the laboratory mill led to the formation of only pure skinnerite (Cu_3_SbS_3_), in the case of the industrial milling, more secondary phases, such as famatinite and tetrahedrite, were present. These phases were most likely formed as side products during the skinnerite synthesis from the elemental precursors. The estimated content of the skinnerite phase in the final product obtained in the industrial mill was around 76%. It is possible that upon prolonged milling in the case of industrial milling, the purity of the product would also improve. Thus, the final sample milled in a laboratory planetary ball mill for 30 min and the final sample treated in an industrial mill for 120 min were selected for further characterization.

### 2.2. Microstructural Characterization of Final Products Prepared in Both Mills

Microstructural characterization was carried out using TEM/HRTEM/SAED/EDX techniques, and the obtained results are displayed in Figure 3. Both samples were made up of small nanocrystalline domains, which aggregated, giving rise to larger particles, as can be observed in the TEM images and the SAED rings.

The synthesized sample in the industrial mill exhibited more-diffuse SAED rings (top inset in Figure 3b), indicating that its crystalline domain was smaller than that obtained for the sample synthesized in the laboratory mill, which is in agreement with the result obtained via SEM, shown later. In the case of the laboratory sample, all of the SAED rings could be indexed in the monoclinic skinnerite (Cu_3_SbS_3_) structure (space group P21/c), in accordance with the XRD results. The (h k l) planes are marked (top inset in Figure 3a). This structure was as well-confirmed via HRTEM as can be seen in the micrograph that corresponds to the [0 2 1]_P21/c_ zone axis. The observed and calculated FFTs are presented in the bottom insets of Figure 3a. The twinning formation can be seen from the HR image (marked with an arrow), probably due to the speedy nucleation in the structure formation. In the case of the industrial sample, the SAED rings (top inset in Figure 3b) were well-indexed to the monoclinic skinnerite (Cu_3_SbS_3_) phase and space group P21/c (in accordance with the XRD). For a monocrystal oriented along the [0 0 1]_P21/c_ phase, the SAED pattern is presented in Figure 3b. A semiquantitative analysis showed that the composition of each sample was very similar to the stoichiometric composition, and the corresponding EDX spectra and atomic percentages are presented in Figure 3.

### 2.3. Morphological Characterization of Final Products Prepared in Both Mills

The morphology of the final samples prepared in the laboratory and in the industrial mill was investigated with SEM, and the representative images are presented in Figure 4. Small particles or crystalline domains that were joined, giving place to larger particles that were about 15 µm in size, were present in both samples. The small domains seemed to be a little larger in the sample obtained in the laboratory-scale planetary mill (100–150 nm) than in the case of the industrial sample (50–100 nm).

### 2.4. Surface Properties of Final Products Prepared in Both Mills

Surface properties were investigated with the nitrogen-adsorption method. The specific surface areas of the final Cu_3_SbS_3_ samples prepared in the laboratory and the industrial mill were 1.3 m^2^g^−1^ and 1.0 m^2^g^−1^, respectively. These low values are in accordance with what can be expected for mechanochemically synthesized ternary chalcogenides [27,28,29]. We could not find relevant information about the BET specific surface area of Cu_3_SbS_3_ in the literature: just the fact that much higher values for the related materials have been reported (e.g., a CuSbS_2_–Cu_3_SbS_4_ nanocomposite prepared with a different method [30]).

To investigate the surface properties of the mechanochemically synthesized Cu_3_SbS_3_ nanocrystals in more detail, the adsorption–desorption isotherms and pore-size distribution were analyzed (Figure 5).

In both cases, the adsorption and desorption sections of the isotherm almost completely overlap, which means the potential amount of mesopores was low. This is in accordance with the low specific-surface-area values described earlier. The shapes of the isotherms at relative pressure values close to 1 hint at mostly macroporous structures. This is confirmed with the pore-size-distribution curves (Figure 5b), namely for the sample milled in the industrial mill (see the high intensities for the pore radii above 20 nm). Although the specific surface areas of both samples were almost similar, it seems that a slightly higher value of the laboratory-milled sample was caused by a slightly larger amount of mesopores and small macropores (the points at low pore sizes exhibited slightly higher intensities in this case, whereas the large macropores were less numerous for this sample). The pore-size distribution calculation from the adsorption section of the isotherm was necessary because when the desorption section of the isotherm of the sample milled in a laboratory-scale ball mill was analysed, an artifact (an intensive peak) at around 2 nm, connected with the tensile strength effect, was detected [31].

### 2.5. Zeta Potential of Final Products Prepared in Both Mills

The zeta potential measurement provided relevant information about the surface and electrostatic properties. The results for mechanochemically synthesized skinnerite (Cu_3_SbS_3_) in the laboratory and the industrial mill, in the pH range from 1 to 12, are shown in Figure 6a and b, respectively. A common feature between both samples was a changeover from positive values, at about pH 1.3, to negative values. The IEP of nonoxidized sulfide minerals is similar to that of elemental sulfur and has been found at pH values between 1 and 2 [32,33,34,35,36,37]. Subsequently, a decrease in ZP values to a level of about −30 mV could be observed in the range of 2 to 4 pH; then, there was an interesting increase in values in the neutral pH range. At alkaline pH, the curve again reached negative ZP values. For the laboratory-milled sample, increasing values passed into the positive area and peaked at pH 8, followed by a decrease in values with a negative maximum of −45 mV at pH 12. With this change, we observed two more isoelectric points, at pH 6.50 and 9.54. These are most likely the result of the presence of the oxidized forms of copper (IEPs: 6.3 pH, Cu(OH)_2_ and 9.5 pH, CuO) [38]. At natural pH 6.05, without addition of ions, the ZP value was −13.8 mV. For the industrial sample, we observed a similar course, but the maximum increase in values reached −17.5 mV at pH 7. If more alkali were added to the suspension, the particles would tend to acquire more negative charge, down to −52.3 mV, for pH 11. The observed increase in zeta potential above pH 11 was caused by compression of the double layer at high ionic forces [38,39,40]. For natural pH 5.2, without addition of ions, the ZP value of this sample reached −32.9 mV. The difference between the ZP values of the samples is most likely a result of different amounts of oxidized forms present at the surface.

On the basis of the results presented so far, only pure skinnerite that had been prepared in the laboratory mill for 30 min was selected to be investigated from optical and optoelectrical points of view.

### 2.6. Optical Properties of Skinnerite Prepared in the Laboratory Mill

The optical properties of the skinnerite nanocrystals were investigated using UV−Vis spectroscopy, which indicated strong absorption in the visible region of the solar spectrum (Figure 7). The inset in Figure 7 shows the Tauc plot that allowed deduction of the direct band gap of the Cu_3_SbS_3_ nanocrystals from the derived UV–Vis spectrum (Figure 7). The optical band gap for the Cu_3_SbS_3_ nanocrystals (Tauc-plot inset of Figure 7) was estimated via plotting (αhν)^2^ as a function of photon energy hν (α = absorption coefficient, h = Planck’s constant and ν = frequency). Extrapolation of a partially linear region, the linear section of the (αhν)^2^ versus hν, allowed an approximate determination of the near-optical bandgap value, estimated to be 1.7 eV for laboratory-prepared Cu_3_SbS_3_, which is somewhat larger than the value (1.5 eV) reported for bulk material [3]. This value is in good agreement with measurements [9] for thin Cu_3_SbS_3_ layers and also for Cu_3_SbS_3_ nanocrystals [7]. The absorption features observed for Cu_3_SbS_3_ nanocrystals are consistent with those of earlier reports [15,41]. The determined value of the bandgap energy shows that this synthesized material can be considered a promising candidate for solar-cell applications.

Raman spectroscopy was employed to inspect phase purity. The structural properties of the synthesized Cu_3_SbS_3_ were also studied using Micro-Raman spectroscopy. The representative micro-Raman spectrum of the sample is shown in Figure 8.

As is evident from the Raman spectrum, the sample showed three main Raman-active modes: at 250, 312 and 344 cm^−1^. The observed Raman modes at 312 and 344 cm^−1^ corresponded to the Cu_3_SbS_3_ phase. The weaker peak at 312 cm^−1^ belonged to the vibrational mode of the Sb-S. The intensive peak at 344 cm^−1^ may be attributed to the phonon vibration modes of the Sb-S_3_ in the Cu_3_SbS_3_. This is in accordance with the results published in paper [6]. The weaker Raman modes observed around 250 and 189 cm^−1^ may be attributed to the Cu–S vibration, which corresponded to that of the CuSbS_2_ in accordance with the literature [42]. The lower Raman modes noticed around 283 cm^−1^, together with those at 312 and 344 cm^−1^, may also correspond to Cu_3_SbS_4_ [43,44,45], as they overlap with the modes corresponding to the Cu_3_SbS_3_ phase. The absence of the CuSbS_2_ and Cu_3_SbS_4_ phases in the XRD pattern of the prepared sample might be due to their existence in small amounts, as the detection limit of the XRD technique is around 5%.

The micro-PL spectrum of the mechanochemically synthesized Cu_3_SbS_3_ is shown in Figure 9. With the excitation wavelength at 514 nm, the corresponding emission peak was found at 647 nm (1.91 eV). Nanocrystalline Cu_3_SbS_3_ emanates a wide range of luminescence in the visible region, depending upon its band gap and size. In the literature, the fundamental band gap of Cu_3_SbS_3_ has been calculated via HSE06 to be 2.02 eV [2], and the experimentally determined optical band gap was 1.84 eV for Cu_3_SbS_3_ film [5] and 1.87 eV of the corresponding nanocrystals [7]. These values are well-suited to the measured PL emission peak in this study. The photoluminescence properties of Cu_3_SbS_3_ have not been studied so far, and only photoluminescence spectra measured with the excitation wavelength at 234 nm and 285 nm [13,14] recorded an experimental bandgap value of 2.95 eV and therefore cannot be compared with our results.

### 2.7. Optoelectrical Properties of Skinnerite Prepared in the Laboratory Mill

Current–voltage (I–V) characteristics were measured to analyze the electrical and optical properties of the Cu_3_SbS_3_ that was mechanochemically synthesized after 30 min of laboratory milling.

As displayed in Figure 10, current–voltage (I–V) curves were measured in a dark state and under white-light illumination with dispersed nanocrystals on interdigital Au contacts. The solution-dipping method allows no preparation of a homogeneously distributed layer of nanocrystalline powder. In this procedure, we assumed that active bridges from nanocrystalline powder would be created in the Au contact gaps of the thinner layers for measurement of electrical and optical properties. The measured characteristic showed formation of conductive bridges with partially nonlinear behavior caused by Au contact. The Cu_3_SbS_3_ nanocrystalline powder of this sample showed a ~45% increase in the photosensitive current at a forward voltage of 0.6 V under illumination in comparison with that in the dark state, which highlights the suitability of this material to be potentially used as the absorber layers in solar cells.

## 3. Materials and Methods

### 3.1. Mechanochemical Synthesis

In a typical working process, the elemental precursors, specifically copper (Merck, Germany), antimony (Merck, Germany) and sulfur (Ites, Slovakia) in a Cu:Sb:S stoichiometric ratio of 3:1:3, were used. In total, 2.33 g of copper, 1.49 g of antimony and 1.18 g of sulfur were loaded into the laboratory planetary ball mill, Pulverisette 6 (Fritsch, Idar-Oberstein, Germany), and milled under the following milling conditions: atmosphere, argon; 50 tungsten carbide milling balls with diameters of 10 mm; mass of milled mixture, 5 g; rotation speed of the planet carrier, 550 min^−1^; milling time, up to 5–30 min. In the case of industrial milling, 46.66 g of copper, 29.80 g of antimony and 23.54 g of sulfur were loaded into the industrial eccentric vibratory mill, ESM-656 0.5 ks (Siebtechnik, Mülheim, Germany). The industrial milling was performed under the following conditions: 5 L steel satellite milling chamber attached to the main corpus of the mill; tungsten carbide balls with a diameter of 35 mm and a total mass of 30 kg; 80% ball filling; amplitude of the vibrations, 20 mm; rotational speed of the eccenter, 960 min^−1^; argon atmosphere. This milling was performed for 60–120 min.

The mechanochemical synthesis was performed according to the following reaction (Equation (1)):*3Cu+Sb+3S → Cu_3_SbS_3_*(1)

### 3.2. Characterization Methods

XRD patterns were collected using a D8 Advance diffractometer (Bruker, Bremen, Germany), with the Cu*K*_α_ radiation in the Bragg–Brentano configuration. The generator was set up at 40 kV and 40 mA. The divergence and receiving slits were 0.3° and 0.1 mm, respectively. The XRD patterns were recorded in the range of 10–70° 2*θ*, with a step of 0.03°. For the phase identification, Diffrac^plus^ Eva and the ICDD PDF2 database were applied, and for the Rietveld analysis, Diffrac^plus^ Topas software was applied.

Morphology and microcharacterization were analyzed using scanning electron microscopy (SEM) and transmission electron microscopy (TEM) techniques. A small quantity of the powder sample was dispersed in acetone, and some drops were deposited on carbon-coated nickel grids (to avoid interference between the Cu grid and the Cu from the sample in the EDS analysis). The SEM images were obtained on a Hitachi S-4800 SEM-Field Emission Gun microscope. The TEM/HRTEM images, the selected-area electron diffraction (SAED) and the energy-dispersive X-ray (EDX) spectra (Oxford Instrument) were taken on a 200 kV JEOL-2100-PLUS microscope (Akishima, Japan) equipped with a LaB_6_ filament (point resolution = 0.25 nm). The HRTEM analysis, the lattice spacing, the Fast Fourier Transform (FFT) and the phase interpretation were carried out with Gatan Digital Micrograph software (Gatan Inc.) and the Java version of Electron Microscope Software (JEM).

The specific surface area was determined with the low-temperature nitrogen adsorption method, using a NOVA 1200e Surface Area & Pore Size Analyzer (Quantachrome Instruments, Boynton Beach, FL, USA). The values were calculated using the BET theory. The complete nitrogen adsorption isotherms were measured in order to determine the pore-size distribution, which was calculated using the Barrett–Joyner–Halenda (BJH) method.

The zeta potential (ZP) was acquired using a Zetasizer Nano ZS (Malvern, Malvern, Great Britain) and obtained from the electrophoretic mobility via the Smoluchowski equation. The ZP was measured in a water solution of 10 mM KCl to maintain a minimum level of conductivity of the medium in the pH range from 1 to 12. These measurements were repeated three times for each sample.

The absorption spectra were taken using an UV–Vis spectrophotometer Helios Gamma (Thermo Electron Corporation, Warwickshire, UK) in the range of 200–800 nm using a 1 cm path length quartz cuvette. The samples were diluted in absolute ethanol with ultrasonic stirring.

The micro-Raman and PL spectra were measured in air at room temperature, with the focus of the beam of an Ar laser (514 and 488 nm) via a confocal Raman Microscope (Spectroscopy & Imaging, Warstein, Germany) in backscattering geometry. The frequency of the Raman line of crystalline Si at 520 cm^−1^ was used to calibrate the system in the present study.

The current–voltage (I–V) characteristics were measured using semiconductor parameter analyzer Agilent 4155C under dark and focused halogen white-light illumination (illumination intensity of ~600 mW/cm^2^). The sample was prepared for this measurement via a solution of nanocrystalline powder in isopropyl alcohol dropped onto an interdigital structure with Au contacts. The interdigital structure area was 3 × 3 mm and the dimensions of the Au finger/gap were 30/12 μm, as shown in Figure 6b. The connection of the interdigital structure to the socket was realized via wires glued with silver paste.

## 4. Conclusions

The mechanochemical approach was successfully applied for the synthesis of skinnerite (Cu_3_SbS_3_) nanocrystals. Monoclinic skinnerite was formed after 30 min in the laboratory mill and after 120 min in the industrial mill. It was concluded that the best sample was that milled in the laboratory mill for 30 min. While the mechanochemical synthesis in the laboratory mill led to the formation of pure skinnerite, in the case of the industrial milling, more secondary phases, such as famatinite and tetrahedrite, were also present. The TEM analysis showed that the nanocrystals were agglomerated into micrometer-sized grains. UV–Vis measurements indicated the suitability of the prepared skinnerite for photovoltaic applications, as the bandgap energy value was 1.7 eV for laboratory milling. Cu_3_SbS_3_ nanocrystals showed ~45% enhancement of the photoresponsive current at a forward voltage of 0.6 V under illumination, indicating the suitability of this material to be potentially used as absorber layers in solar cells. Application of the industrial eccentric vibratory milling for skinnerite synthesis and its large-scale production reported here might be interesting for researchers in the field of photovoltaics.

## Figures and Tables

**Figure 1 molecules-28-00326-f001:**
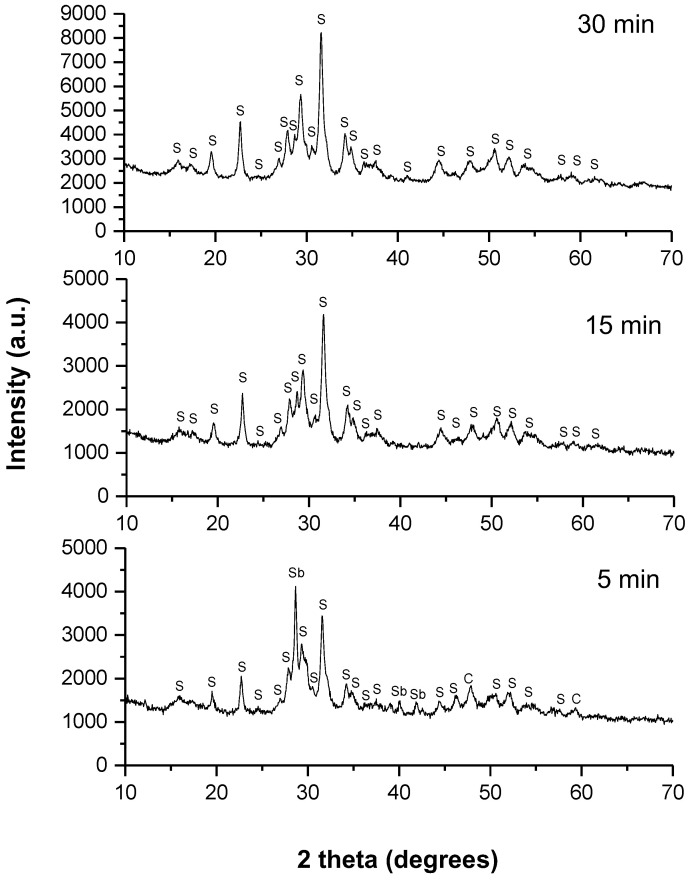
XRD patterns of the milled mixture of Cu, Sb and S in a stoichiometric ratio for different milling times in the planetary mill. The identified phases are marked as follows: Sb—unreacted antimony, CuS—covellite (C), Cu_3_SbS_3_—skinnerite (S).

**Figure 2 molecules-28-00326-f002:**
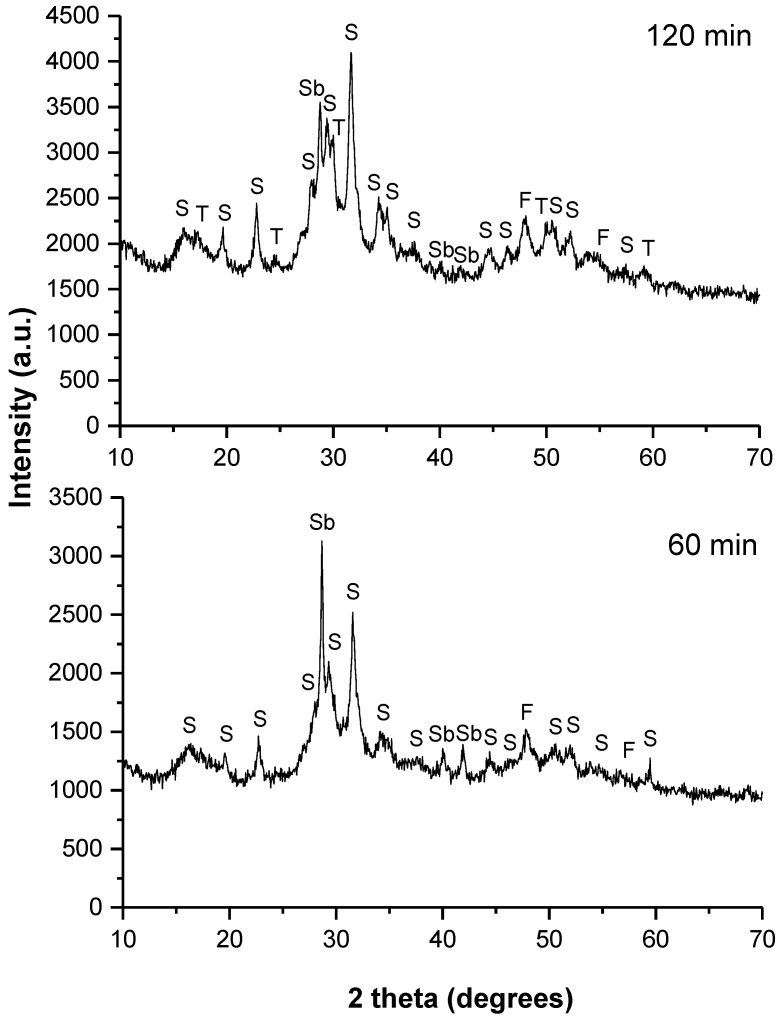
XRD patterns of the milled mixture of Cu, Sb and S in a stoichiometric ratio for different milling times in the industrial mill. The identified phases are marked as follows: Cu_11,4_Sb_4_S_13_—tetrahedrite (T), Cu_3_SbS_4_—famatinite (F), Cu_3_SbS_3_—skinnerite (S) and Sb—unreacted antimony. The labeled phases are only informative due to overlap of some phases.

**Figure 3 molecules-28-00326-f003:**
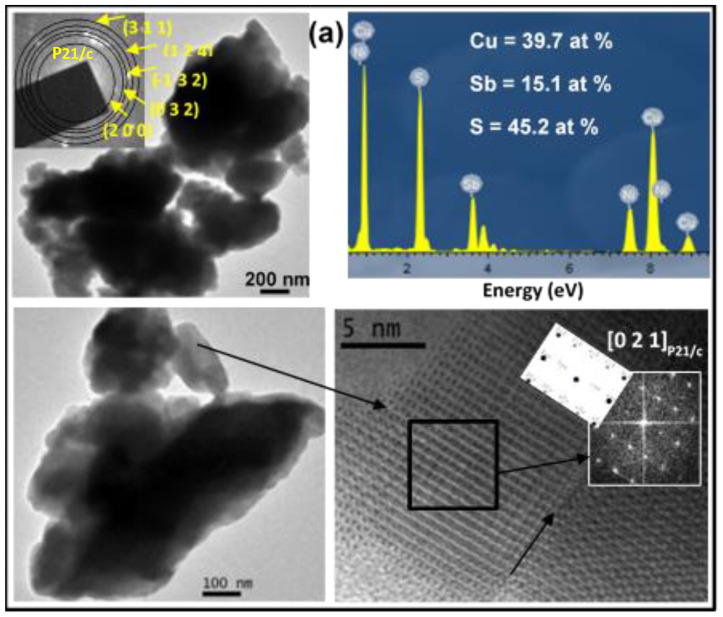
TEM/HRTEM images, SAED rings and patterns, and FFT and EDX spectra of the final Cu_3_SbS_3_ products synthesized (**a**) in the laboratory and (**b**) in the industrial mill.

**Figure 4 molecules-28-00326-f004:**
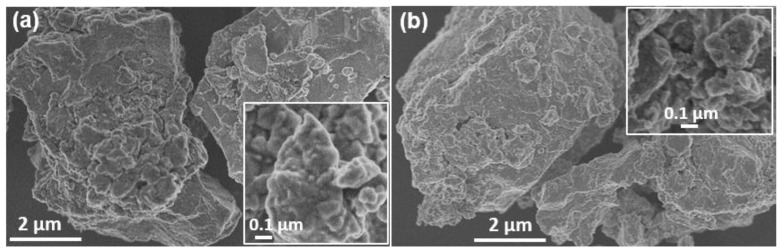
SEM micrographs of mechanochemically synthesized Cu_3_SbS_3_ in (**a**) the planetary mill and (**b**) the industrial mill. The insets correspond to higher magnifications.

**Figure 5 molecules-28-00326-f005:**
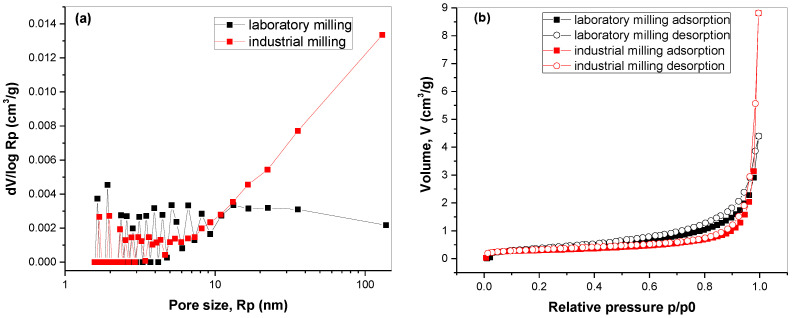
(**a**) Adsorption and desorption isotherms and (**b**) pore-size distributions calculated from part of the adsorption isotherm for mechanochemically synthesized Cu_3_SbS_3_ in the laboratory and the industrial mill.

**Figure 6 molecules-28-00326-f006:**
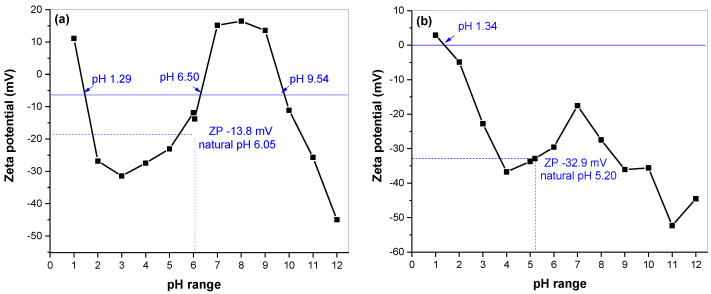
Dependence of zeta-potential values on pH for mechanochemically synthesized Cu_3_SbS_3_ in the laboratory (**a**) and the industrial mill (**b**).

**Figure 7 molecules-28-00326-f007:**
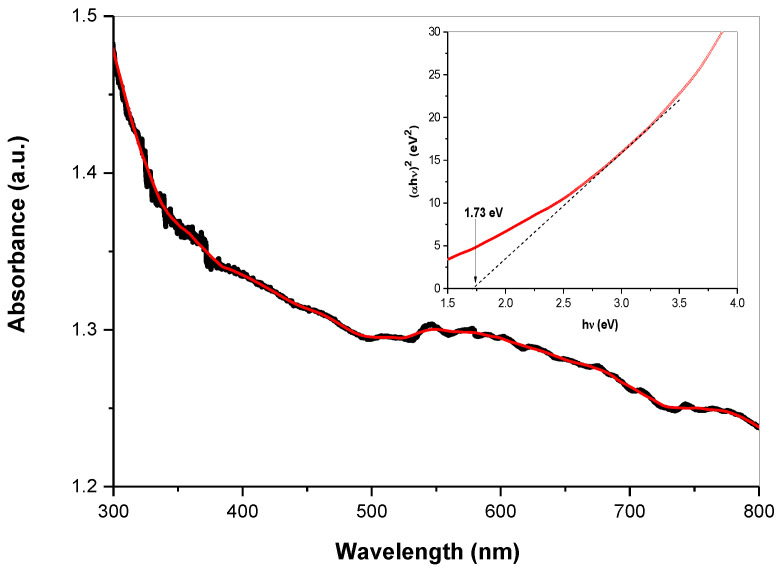
UV–Vis spectrum and Tauc plot with the determined optical energy band gap (inset) of mechanochemically synthesized skinnerite in the laboratory mill. Red line—smoothed spectrum.

**Figure 8 molecules-28-00326-f008:**
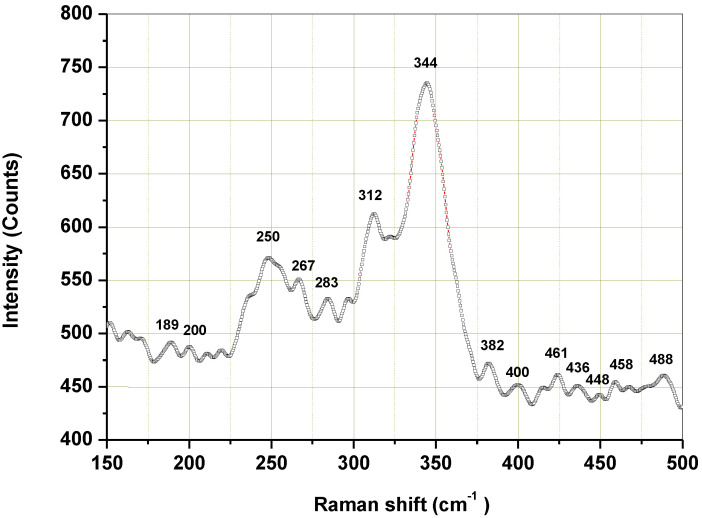
Micro-Raman spectrum of the Cu_3_SbS_3_ nanocrystals synthesized in the laboratory mill.

**Figure 9 molecules-28-00326-f009:**
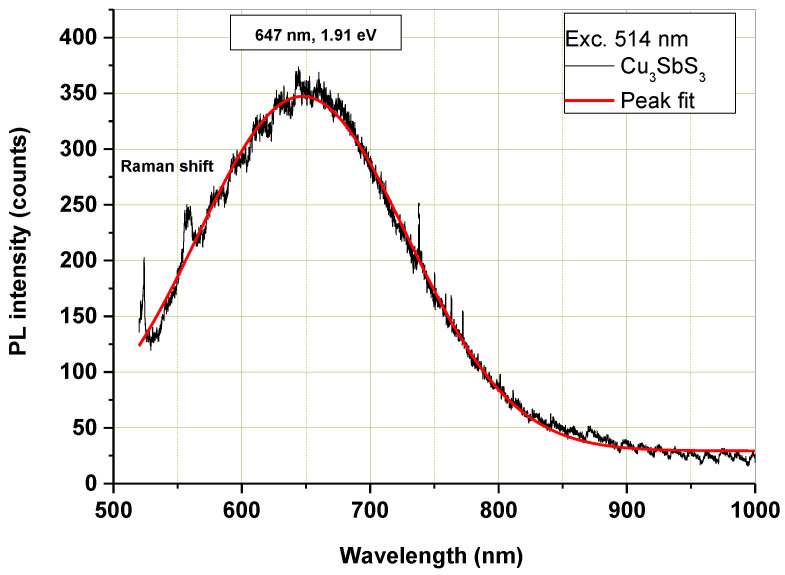
Microphotoluminescence spectrum of mechanochemically synthesized Cu_3_SbS_3_ in the laboratory mill.

**Figure 10 molecules-28-00326-f010:**
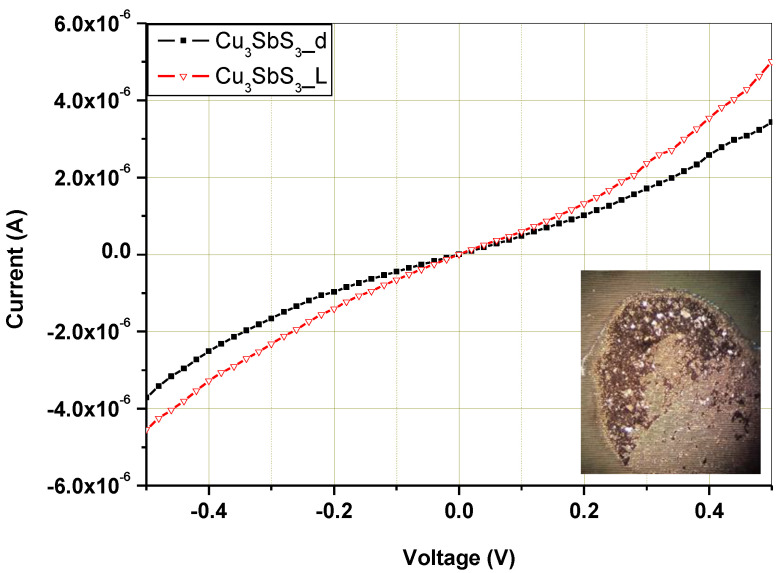
Current vs. voltage (I–V) characteristics of mechanochemically synthesized Cu_3_SbS_3_ in the laboratory mill in a dark state and under light illumination, and an optical image of dispersed nanocrystals on interdigital Au contacts (inset).

## Data Availability

The data presented in this study are available on request from the authors.

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
