# Peer review of "Nanocrystalline Skinnerite (Cu3SbS3) Prepared by High-Energy Milling in a Laboratory and an Industrial Mill and Its Optical and Optoelectrical Properties"

_molecules, 2022, doi:10.3390/molecules28010326_

Round 1

Reviewer 1 Report

Production of skinnerite either in laboratory and industrial mill is demonstrated and proposed as an interesting way to produce Cu3SbSe3 for photovoltaic applications. If investigation of structural and morphological properties gives significant results, in my opinion the optical and optoelectrical characterization is weak and not so convincing. I believe necessary a major revision of this last part of work before publication, in according with the following suggestions:

Fig. 1 show that a longer milling time, from 15 to 30 minutes, increases XRD intensity. Is that a sign of improvement in the crystalline quality? Longer times could give better results or 30 minutes is the optimal time?

Fig. 3:  some labels are too small.

Paragraph 2.4: How are surface properties with respect ones of samples prepared from other synthesis techniques?

Fig. 7: the use of the Tauc plot to extrapolate the band gap width is not quite adequate for me in this case. Absorbance is low and the absorption edge is not so sharp to have reasonable result from the Tauc plot. This is evident in the inset, in which no evident linear trend is present and the tracing of the line to identify the band gap is absolutely arbitrary. Probably, it is possible to measure the absorbance in a more concentrated solution or by preparing a solid tablet in KBr. In the absence of a more convincing measure, I would avoid to declare a band gap around 1,7 eV. In fact, PL measurements also give slightly different results.

Fig. 10: collection of photocarriers is strongly dependent on the contact geometry. What is the pattern of the interdigital Au contacts? What is the thickness of deposited materials? In the picture of inset the sample is highly reflecting and the surface is very roughness. In the picture of inset the sample is highly reflecting and the surface is very roughness. This could be a problem ...

Author Response

We thank to the reviewer for favorable comments and suggestions. We incorporated corresponding changes into body of the manuscript and provided the responses to the comments.

Reviewer 2 Report

The main point of interest and novelty of the work is the perspective of production and synthesis of skinnerite at industrial scale with good properties.

For that purpose, the authors compare the properties of the systems obtained by laboratory and industrial mills.

The production and synthesis by using laboratory mill is shown to be successful for 30 min milling times the single phase obtained is skinnerite (S).

However, the authors could not reproduce the same good results using the industrial milling with milling times 0f 60 min and 120 min.

Meaning, it was not possible, using industrial milling to get the product as pure (only S) as that obtained with the laboratory milling.

It is not clear why, upon such result obtained with industrial milling with 60 and 120 min, it was not done the production with industrial mill for shorter times, for example 30 min. Since, apparently, shorter times gives better results in the sense of obtaining a single phase, Skinnerite .

Author Response

We thank to the reviewer for favorable comments. We provided responses to the comments of the reviewer.

Reviewer 3 Report

The presented manuscript describes preparation by mechanochemical technique and characterization of Cu3SbS3. Samples were prepared in laboratory and industrial mills from a stoichiometric mixture of elements.

Obtained samples were characterized by different experimental methods, and it was shown that laboratory mill prepared sample is more phase pure and interesting for photovoltaic applications.

Low cost and relatively non-toxic materials are in the focus of material science due to the possible application to develop renewable energy sources by photovoltaic or thermoelectric devices. From this point of view, the study of preparation of Cu-Sb-S compounds is in state of the art.

Some questions arose during manuscript reading.

a) The powder XRD patterns shown on Figs 1 and 2 contain some reflexes that were not described in the figure legends. What is Sb on figs 1 and 2? And I could not find CH on Fig 2. If you compare patterns from Fig 1 “5 min” and Fig 2 “60 min” it is possible to see the similarities. But the sharpest reflex, for example, was attributed on Fig 1 to CuSbS2 (CH) but on Fig 2 to “Sb”

b) Is it possible to estimate crystallite size from powder XRD and compare with the value obtained by microscopy?

c) I did not understand what was the reason to determine zeta potential for prepared samples. These results were not used for following optical characteristics. And what data may be expected from industrial mill produced sample zeta potential measurement containing only 76 % of target compound? Is it nor clear that more phase pure laboratory mill prepared sample is more suitable for optical and optoelectrical properties investigation?

Author Response

Many thanks to the Reviewer for his comments. 

Round 2

Reviewer 3 Report

I agree with corrections made in the text of the manuscript.  The zeta potential measurement justifications that were in the answers on reviewer's comments may be included into the text